# Be Prosocial My Friend: The Social Disconnection Model of Perfectionism in Adolescents Immersed in Competitive Sport

**DOI:** 10.3390/ijerph20042887

**Published:** 2023-02-07

**Authors:** Alvaro Rodríguez-Franco, Gustavo Carlo, Pedro Valdivia-Moral, Juan González-Hernández

**Affiliations:** 1Department of Personality, Evaluation and Psychological Treatment, Faculty of Psychology, Campus Cartuja, University of Granada, 18071 Granada, Spain; 2Cultural Resiliency and Learning Center Equity Advisor, School of Education, University of California, Irvine. 401 Peltason Drive Suite 3453, Irvine, CA 92617, USA; 3Department of Musical, Plastic and Corporal Expression Didactics, Faculty of Educational Sciences, Campus Cartuja, 18071 Granada, Spain

**Keywords:** prosocial behavior, aggressiveness, competitive contexts, social relationships, perfectionism

## Abstract

The aim of the present study is to explore the relationships between cognitive–behavioral patterns of perfectionism in the context of competitive sport and both prosociality and aggressiveness in a sample of adolescents competing in federated sports. A cross-sectional and non-randomized study was designed using a selective methodology on a sample of adolescents (N = 234) competing in federated sports. Scales to assess aggressiveness, perfectionism, prosocial behavior, and competitiveness were administrated. The results show that as age increases, prosocial behaviors increase and aggressive behaviors and competitiveness decrease, while there was no one significant perfectionist tendency. Competitiveness showed a direct relationship with aggressive (positive) and prosocial behaviors (negative). Self-oriented perfectionistic behavior showed a direct and significant relationship with prosocial behaviors, but no significant relationship with aggressive responses. As P-SP and P-OD tendencies increased, significantly smaller links were shown with prosocial behaviors, but greater links were shown with aggressive behaviors. A path (mediation) model showed a positive and predictive relationship with aggressive behaviors and a negative relationship with prosocial altruistic behaviors. The negative influence of criticism from significant figures in their environment and unrealistic expectations about their performance are relevant to difficulties in self-regulating social relationships in adolescents. Hence, it is a challenge to promote prosocial resources (as a protective value for aggressive behaviors) in the face of the early angst of young athletes, who put their maturity to the test under conditions of high pressure and demands. The present study continues to reinforce the line drawn on perfectionism and prosocial development in young people in sports contexts where young people, measured early on according to their performance, can accentuate and deepen competitive tendencies that alter their adaptive and self-regulatory capacities, as well as their psychosocial projection.

## 1. Introduction

A context that measures the individual by their performance is characterized by strong social pressure [1], a fixation on sports goals/outcomes by which to generate worth [2], and the development of beliefs that failure to achieve them is socially detrimental (e.g., failure) [3]. Youth competitive sports is one prime example of this context. Faced with the need to learn the practice of a sport modality, with its rhythms and progressions, a young athlete is subjected to the influence of expectations (e.g., of family members, coaches) and those self-imposed behavioral and/or aspirational standards observed in their psychosocial interactions (e.g., oriented toward winning in sport, pleasing others) [4,5]. Positive psychosocial adaptation is nurtured by adaptive processes (e.g., moral attitudes) that can emerge from a supportive and developmentally appropriate context (e.g., team sports, individual sports) created by the feedback and expectations of coaches and other significant figures (e.g., families, referees) towards the behaviors of young athletes [6].

However, when family members and/or coaches do not provide support or consider age-related, maturational developmental processes [7], a young athlete may be unable to differentiate between constructive and destructive criticism, making it even more challenging to adeptly manage such feedback [8]. In this latter context, socially inappropriate negative attitudes towards others (e.g., arrogance, low empathy) and the emergence of aggressive and maladaptive responses [9,10,11] can manifest, and such consequences can endure into adult life [12]. Indeed, researchers have shown that overly competitive sport at immature ages with a focus on excessive ego orientation, excessively controlling styles by coaches, controlled motivation, and a need for status, is associated with the emergence of high moral disengagement in young athletes [13,14].

### Perfectionism, Achievement Orientation, and Social Relationships

Multidimensional models of perfectionism have been proposed in the past decade [15,16,17]. According to Stoeber et al. [18], perfectionism has been related (Figure 1) to functional/dysfunctional responses toward achievement in interpersonal relationships. For example, socially prescribed perfectionism (PS-P) has been related to the development of beliefs rooted in the context of reference (e.g., being the best in sports) and in the search for perfection as a standard of social excellence. Self-oriented perfectionism (P-SO) has been shown to relate to adaptive factors such as the pursuit of order and discipline in sports, responsibility, and organization (e.g., striving to find a better version of yourself) [19], whereas others-oriented perfectionism (P-OD) focuses on modifying (even forcibly) the context to successfully meet achievement standards.

Perfectionism, which is linked to achievement orientations, arises in performance contexts and drives individuals to attempt to stand out, excel, and seek perfection. On an interpersonal level, perfectionism affects the way we relate and connect with people. According to the perfectionism social disconnection model (SDPM; [20,21]) (Figure 1), the social interactions of maladaptive perfectionists generate an experience of both objective and subjective social disconnection. Perfectionism, specifically in adolescents, has a negative impact on their psychological well-being [22,23,24].

The World Health Organization [25] emphasizes the danger of not addressing mental disorders in early developmental stages due to the subsequent repercussions on physical and mental well-being in adolescence and adulthood. Hawkins et al. [26] showed that between 25% and 30% of young people present traits of maladaptive perfectionism. In recent studies, investigators have found that for every ten adolescents, three manifest maladaptive perfectionism [27,28]. Such social disconnection can be observed in withdrawal from others, conflicting relationships caused by competitiveness, a lack of companionship in the classroom or sports club, infrequent social contact, and poor social support. Most perfectionist children experience high levels of sensitivity and consequently experience fear of criticism, insecurity in relationships, a need to be accepted and admired, a need to please others, and a need to base their self-concept on the valuation of others, all of which are linked to feelings of rejection and/or exclusion [29].

Although there is no research on perfectionist social disconnection in young people who practice competitive sports, it is a model that could explain the psychological experience of young people subjected to high pressure in sport at an early age. Rice et al. [30] differentiated a sample of 984 adolescents into three groups with different perfectionist tendencies—adaptive, maladaptive, and non-perfectionists—and found that both adaptive and maladaptive perfectionists exhibited more prosocial behaviors and maintained better interpersonal relationships than non-perfectionists, as corroborated by both parents and peers. However, they observed that maladaptive perfectionists obtained less social acceptance and popularity among their peers. Furthermore, in comparison with adaptive perfectionists, maladaptive perfectionists engaged in greater self-criticism and showed less willingness to participate in activities (unless success was assured), as well as more self-limiting behaviors.

Aggressiveness (harmful or injurious actions, sometimes enhanced by excessive competitiveness) is a derivation of the anxiety mechanism because it presents a reflexive emotional response (e.g., frustration) to not being able to handle or control a situation [31]. It is not surprising then that in youth sports, more perfectionistic tendencies lead to higher levels of frustration (e.g., low standards for achieving excellence), which gives rise to feelings of helplessness and incompatibility that can lead to anxiety-aggressiveness [32]. Studies have shown the existence of positive correlations between others-oriented perfectionism and socially prescribed perfectionism and negative correlations between others-oriented perfectionism and self-oriented perfectionism, and even a possible negative correlation between the latter and aggressiveness [12,32]. According to the postulates of the perfectionism social disconnection model, aggressiveness emerges as a negative response to imperfection [33], though it differentiates between behavioral problems in boys (e.g., aggressiveness, hostility, inattention, opposition to parents and teachers) and emotional problems in girls (e.g., anxiety, distress or sadness) [34].

Participation in organized sports offers young athletes a wealth of opportunities for social interaction (e.g., with peers and adults) [35] which can enhance their maturation and instill moral norms and values [36]. Challenging sports activities and cooperative team sport actions often involve actively facing moral decisions (e.g., fair play, cheating) [37,38]. Prosocial behaviors are actions intended to benefit one or more persons. When observing sport, it is possible to observe selfish behaviors when participants seek to increase only their own well-being (e.g., by not passing the ball, wanting to win at all costs), whereas, since the goal of prosocial tendencies is to increase the well-being of other people, they lead to the development of altruistic behaviors (e.g., accepting that a ball has gone out without complaining to the referee) [39,40], empathic behaviors (e.g., understanding the efforts of rivals and teammates) [41], or cooperative behaviors (e.g., feeling supported by teammates to achieve a common goal) [42].

The aim of the present study is to explore the relationships between cognitive–behavioral patterns of perfectionism in the context of competitive sport and both prosociality and aggressiveness in a sample of adolescents competing in federated sports. To this end, and accounting for gender, the hypothetical model (Figure 2) suggests that: (1) adolescents in competitive contexts will show positive perfectionist tendencies toward aggressive tendencies, but negative tendencies toward prosocial behavior, and (2) prosocial behavior will exert a protective influence on the relation between perfectionism and aggressive tendencies in young athletes.

## 2. Materials and Methods

### 2.1. Sample, Procedure, and Data Collection

A sample of 234 adolescents (M = 16.1_years_; DE = 2.96) from different sports clubs and federations participated in the study. A descriptive analysis yielded evidence that there were more boys than girls in terms of the proportion federated in their sport (♂54.25% vs. ♀26.43%) and that boys reported higher feelings of competitiveness in sport than girls (♂64.31% vs. ♀41.36). Boys were typically more involved in the practice of collective sports than girls (♂47.87% vs. ♀19.28), while a higher proportion of girls described themselves as “not at all competitive” (♀35% vs. ♂19.14%) and as having a higher feeling of competitiveness in academics (♀40.42% vs. ♂ 12.14%).

A cross-sectional, non-randomized, relational study was designed using a selective methodology and a survey was designed with a questionnaire format. Prior to data collection, permission was requested from both the legal guardians (parents) and the center (management teams), who were also informed of the objectives of the study, the voluntary nature of participation, and the fact that participants had absolute freedom to leave the study at any time. In addition, the informed consent form included information concerning the form of contact with the researchers and their commitment to confidentiality and the anonymity of the participants, as well as the methodological rigor that would be applied to the information provided. An online Google Form application was developed with response templates for the evaluation instruments. Teaching tablets or smartphones were used to administer the questionnaires. The questionnaires were administered at a single designated time in the classroom, without the presence of the teaching staff and in the presence of the research staff, guaranteeing the anonymity and confidentiality of the data obtained. The latter could answer any questions that arose during the completion of the questionnaires. The questionnaires and their administration protocol complied with the provisions of the University of Granada Ethics Committee (1726/CEIH/2020).

### 2.2. Measures

Sociodemographics. An ad-hoc self-report form was designed with the aim of answering some general questions about the sample collected (e.g., age, sex, level of study, academic degree, federated status, subjective competitiveness perception (SCP)).

Physical and verbal aggression. For the measurement of this variable, we used the Spanish adaptation of the Physical and Verbal Aggression Questionnaire (AFV) [43], which is composed of 20 items concerning typical situations that may occur in daily life, 5 of which function as controls that are not computed in the results. These items describe both physical and verbally aggressive behaviors, accompanied by a Likert-type graduated scale in three levels of frequency: (3) “often”; (2) “sometimes”; (1) “never”. The scale provides a total aggression score and three factor scores for physical aggression, verbal aggression, and control of aggressive behavior. The internal consistency of the questionnaire has been shown to be high, with a Cronbach’s alpha of 0.79, and the confirmatory analysis (CFA) maintains the one-dimensionality of the original version (χ^2^/gl = 29.78; *p* = 0.00; CFI = 0.93; NNFI = 0.92; CFI = 0.92; SRMR = 0.08; RMSEA = 0.03).

Perfectionism. The Spanish version of the Multidimensional Perfectionism Scale (MPS) by [16], adapted by [44], is a measurement instrument composed of 45 items that describe three subscales or essential components of perfectionist behavior: (a) self-oriented perfectionism (P-AO; “I always demand perfection from myself”); (b) socially prescribed perfectionism (P-SP; “I have difficulty meeting the expectations that others have of me”), and (c) others-oriented perfectionism (P-OD; “I rarely criticize my friends, when they conform they do so with low quality”). The questionnaire consists of a Likert-type scale with seven response options referring to personal characteristics or traits, where value 1 represents total disagreement and value 7 represents total agreement. The scale shows an internal consistency of 0.86 while the CFA shows an adequate fit (χ^2^/gl = 32.45; *p* = 0.00; CFI = 0.90; NNFI = 0.92; CFI = 0.94; SRMR = 0.07; RMSEA = 0.04).

Prosocial Behavior. The Adolescent Prosocial Tendencies Scale [45] was applied. We used the altruistic prosocial behavior scale, composed of 21 items, whose responses are collected on a Likert-type scale with values of 1 (“I do not identify myself”), 2 (“I identify myself little”), 3 (“I identify myself somewhat”), 4 (“I identify myself well”), and 5 (“I identify myself very well”). The internal consistency of the questionnaire has been shown to be good, with a Cronbach’s alpha of 0.70, and it shows an adequate fit (χ^2^/gl = 23.17; *p* = 0.01; CFI = 0.91; NNFI = 0.92; CFI = 0.96; SRMR = 0.08; RMSEA = 0.03).

### 2.3. Data Analysis

Initially, a descriptive analysis of the sample (mean, standard deviation, minimum and maximum) and consistency of measurement tests (Cronbach, CFA, d Cohen, and maximum likelihood) were performed [46]. The normality of variances was calculated (K-S; >0.05), verifying that all the variables analyzed conformed with a normal and parametric distribution, and *t*-tests of mean differences (by gender) were performed. The Pearson correlation showed a linear relationship (*p* < 0.05) between all the variables under study. A mediation analysis of the observed variables was performed using the maximum likelihood estimation method. A model was run to test the direct path between patterns of perfectionism, prosocial behavior, and aggressive tendencies. Error variances were allowed to correlate with each other, and the indirect pathway between patterns of perfectionism, and aggressive behavior through prosocial behavior was tested. The effect of sex, subjective competitiveness perception, and age was controlled for to test the robustness of the results. Finally, a multi-group analysis was performed to examine differences in the hypothesized model according to gender. An incremental chi-square test and an incremental NFI test were performed to examine whether there was a significant change between a restricted and an unrestricted model for different genders [47]. Statistical analyses were conducted using the Statistical Package for the Social Sciences SPSS 25 and PROCESS version 26 [48,49].

## 3. Results

Table 1 shows the differential and invariance analysis. The subjective competitiveness perception served to indicate the subjective response to the influence of competition in young athletes. With this, and with the great majority of the sample collected showing certain tendencies towards competitiveness, Student’s *t*-test analyses by gender did not show significantly higher means in any perfectionistic tendency but did show significantly higher means in altruistic prosociality in girls (<0.05). In contrast, boys showed significantly higher differences in subjective competitiveness perception relative to girls (=0.00). The multi-group analysis showed non-significant differences between the model without constrictions and the model with constrictions χ^2^ (6, N = 227) = 6.21 (n.s). The first model, with constrictions, assumes that all relationships between variables are equal for boys and girls, whereas in the second model, without constrictions, all coefficients are estimated in both groups. Since there are no differences between them, the model with more degrees of freedom is the most appropriate. This result supports the structural invariance of the model in both genders, which increases the generalizability and applicability of the model.

Correlation analyses (Table 2) showed linear relationships that help to explain the most significant links between the variables. In this sense, it can be observed that as age increases, prosocial behaviors increase (>0.05) and aggressive behaviors (=0.00) and subjective competitiveness perception (<0.01) decrease, while there is no significant change in perfectionist tendencies. A significant and direct relationship appears to exist between subjective competitiveness perception and aggressive behaviors (<0.01), while it was the inverse with prosocial behaviors (<0.01). As for perfectionistic tendencies, a direct and significant relationship between self-oriented perfectionism and prosocial behaviors (<0.05) can be seen in young athletes, though neither exhibited a significant relationship with aggressive responses. In other words, as P-SP and others-oriented perfectionism tendencies were increased, significantly smaller links were shown with prosocial behaviors (<0.05), and greater links were shown with aggressive behaviors (=0.00).

Once the non-existence of differences was verified and the covariances were restricted so that they were equalized according to gender, and taking into account the higher degrees of significance between the variables in the correlation analysis, different estimations were performed to reach a path analysis from perfectionistic tendencies to aggressive behaviors (i.e., intimidation, insults, aggressions), and this provided a good fit with the data (X^2^ (df = 36) = 61.74; *p* = 0.04, CFI = 0.991, RMSEA = 0.039). In both the first model (for boys) (differential χ^2^ (df = 21) = 7.82; *p* < 0.01) and the second (for girls) (differential χ^2^ (df = 32) = 10.37; *p* < 0.02), significantly adequate fits were achieved.

Figure 3 shows that the path (mediation) model showed that while socially prescribed perfectionism (*p* = 0.00) and aggressive behaviors (*p* < 0.01) showed a predictive and positive relation, altruistic prosocial behaviors exerted a protective value for aggressive behaviors in young athletes (*p* = 0.00).

## 4. Discussion

The contingencies that govern and operate in youth behavior depend on the interaction of the person with his or her environment, and this is important to consider in order to better understand how persons will act in a specific context [49]. Federated sport, characterized by competitiveness, is a context that is characterized by measuring an individual by his or her performance (e.g., sports results, ranking). The hypotheses proposed, while accounting for gender differences, sought to show evidence that adolescents in competitive contexts would show perfectionist tendencies towards more aggressive tendencies, and negative tendencies towards prosocial behavior (H_1_), while prosocial behavior would exert a protective influence on the relation between perfectionism and aggressive tendencies in young athletes (H_2_). As expected, the present findings suggest that a perfectionist orientation (mainly those patterns referred to as socially prescribed) can lead to more conflicts and difficulties in social relationships in contexts where there is a high perception of competitiveness. Importantly, the findings also yield evidence that altruistic prosocial behaviors might attenuate the link between perfectionism and aggression.

The main findings suggest that altruistic prosocial behaviors significantly weakened the relation between socially prescribed perfectionism and aggressive behaviors. It is precisely socially prescribed perfectionism—the most complicated perfectionist orientation to interpret [50] and the one that characterizes people under intense pressure (perceived or real) from others who expect and demand perfection and who feel pressure to meet the extreme expectations of demanding people or the surrounding context—that showed this protective effect. The socially prescribed perfectionist athlete is sensitive (and vulnerable because of his or her maladaptive condition) to perceived external demands for perfection from family members, coaches, and peers, or from the sporting context in general, giving meaning and logic to the pressure [5,21,51,52,53]. It is probably the dark side of perfectionism, discussed by [54] and endorsed by [29,55], that is linked to resentment, anger, and hostility, and it is one of the most maladaptive behavioral responses derived from the pressure to perform, to want to be better than and superior to others. It is in this sense that prosocial traits, as a reflection of bonds built through helping behaviors and emotional, moral, or decisional affinity (e.g., working together towards common goals) with peers or superiors (e.g., coaches), become a protective factor of the aggressive response when athletes are subjected to high pressure due to competitive sports [56,57,58].

Prior research shows that behaving prosocially is significantly related to achievement tendencies [52,56,59]. It is also worth noting that other studies have shown that adaptive perfectionism is mediated by the practice of physical activity and adolescents’ goals in federated sports [60], where sport intensity and social goals serve as personal standards for the good psychosocial development of the adolescent [61]. Given the relations between perfectionism and prosocial behaviors, the present findings thus imply that perfectionism could have long-term implications for the academic outcomes of young people.

There were some interesting gender-related findings. In the present study, regarding socially prescribed and other-oriented forms of perfectionism, there were no significant differences between boys and girls, which is consistent with prior research conducted in other performance contexts (e.g., school) [17,62,63]. These findings are also consistent with those of previous studies that report differences between boys and girls in non-competitive, non-sports samples [21,22,64]. Moreover, and in line with previous scientific findings [35,52,59], as competitiveness in their achievement trajectories intensifies, aggressive tendencies also increase, but prosocial tendencies decrease in both boys and girls. However, studies based on the perfectionism social disconnection model [20,21,33] point out that girls who exhibit self-oriented perfectionism tend towards higher levels of agreeableness and gregariousness [33,42]. Taken together, these findings suggest the need for more research examining these relations across genders to better ascertain whether specific forms of perfectionism are associated with distinct developmental outcomes for boys and girls.

Interestingly, both the positive relationship between competitiveness and aggressive tendencies and the negative relationship between competitiveness and prosocial tendencies alleviate doubts raised in earlier research [65,66,67,68]. Bruner et al. [37] observed in a study with similar samples that prosociality is a greater discriminating factor in predicting developmental outcomes in adolescents who participate in federated sports than in those who do not. However, in the same study, they also showed that greater socialization difficulties (e.g., impulsivities) acted as a mechanism that can inhibit prosociality [37,69,70]. As noted previously, prosociality is linked to positive academic outcomes [71,72]. Following this line of work, the present findings suggest that strong competitiveness might mitigate prosocial behaviors, and indirectly inhibit academic achievement. Interestingly, other studies have shown that adaptive perfectionism is mediated by physical activity and goals in samples of adolescents who participate federated sports [23,57,69], and that sport intensity and social goals serve as personal standards for adolescents’ positive psychosocial development.

### Limitations and Future Research Proposals

It is necessary to point out some of the limitations of this study. The sample size, although relevant, requires us to be scrupulous when making generalizations based on the data, or when drawing comparisons with other samples in which different types of sports contexts are considered (for example, team sports versus individual sports). It is also important to maintain the precautions derived from the cross-sectional condition of the study’s design, as well as the contextualization of young people in different conditions of sporting competitiveness (for example, federated only in regional competitions). Finally, regarding permits and access to athletes, difficulties involving contact, bureaucratic delays, and waiting times due to ethical protocols (for example, access to family permits) were experienced. Future research should continue to explore the relations between these variables and involve study participants from other regions and countries, and from other sports modalities. Stronger inferences concerning causality and the direction of effects would be possible with stronger study designs (e.g., intervention, longitudinal), and this would enable us to better establish evidence concerning the impact of perfectionist orientations on prosocial and aggressive behaviors in competitive sports contexts.

## 5. Conclusions

The present study contributes to our understanding of the relations between perfectionism and prosocial and aggressive development in young people in sports contexts. The perfectionist orientation in competitive sports contexts is inevitable; it is inherent to the act of surpassing oneself and achieving sporting goals. Hence, it is important to prepare young athletes (and especially those who will be subjected to greater sporting and social pressures) to cope with and manage both their emotional and behavioral processes and social relationships in sporting contexts. In young players, the criticisms of significant figures in their environment or the creation of unrealistic expectations concerning their performance (more maladaptive orientations of perfectionism) can create contexts of unhealthy rivalry, endangering the generation of prosocial bonds at the cost of winning or not losing. It is in these emotional states that the difficulty to self-regulate in social relationships will be greater, and in which behaviors intend to satisfy the demands and alleviate the discomfort of sporting pressure will be observed.

Some practical implications can be pointed out for sports professionals involved in the training of young athletes. First, providing positive support to young athletes can contribute to the development of individual traits that allow them to adapt to the demands of the competitive context and develop positive psychosocial skills for the enhancement of their sports careers. In addition, it is essential that the adults (e.g., coaches, parents) surrounding young athletes pay attention and undertake efforts to improve their involvement during training and matches, regulating and projecting demands that favor the sporting development of their children/athletes.

Promoting pro-social resources is a challenge in the face of the early angst of young athletes who put their maturity to the test under conditions of high pressure and demands. These strategies, if internalized and incorporated by coaches and parents, but also by the young athletes themselves, will favor adequate motivational climates, positive transformation through adult leadership, the incorporation of reasonableness in the work of understanding sports rivalry, and socioemotional improvements in competing and overcoming challenges in sports training, as well as the psychological adaptation of young athletes.

## Figures and Tables

**Figure 1 ijerph-20-02887-f001:**
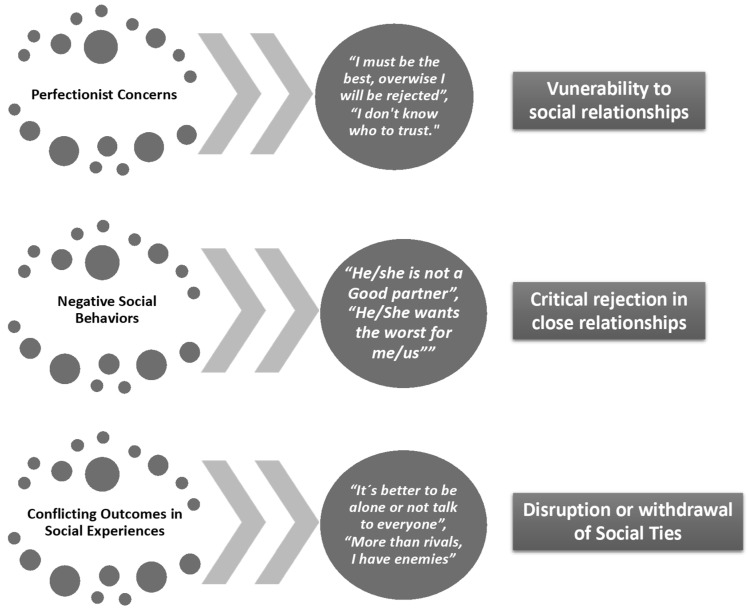
The perfectionism social disconnection model (adapted by the authors from [20]).

**Figure 2 ijerph-20-02887-f002:**
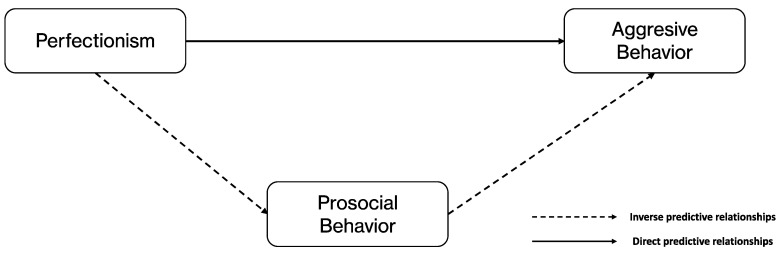
Hypothetical perfectionism–social relationship model for youth sport.

**Figure 3 ijerph-20-02887-f003:**
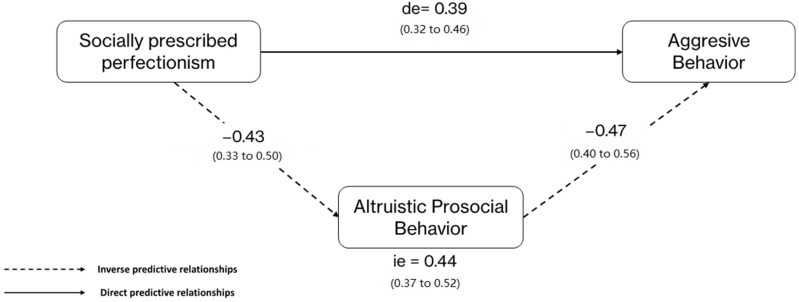
Perfectionism–social relationship model for youth sport.

**Table 1 ijerph-20-02887-t001:** Multi-group analysis of the model by gender.

			SCP		Frequency (%)			Girls n (%)		Boys n (%)
			Not competitive		67 (28.63%)			49 (35)		18 (19.14)
			Somewhat competitive		49 (20.94%)			32 (22.85)		17 (18.08)
			Quite competitive		63 (26.92%)			42 (30)		21 (22.34)
			Very competitive		55 (23.50%)			17 (12.14)		38 (40.42)
**N = 234** [(girls = 146; boys = 88; gl = 232). (K-S = 0.54 (0.26)]								**t**	** *p* **	**d**
**P-SO** [(girls; $\bar{X}$ = 4.62; DE = 1.06); (boys = 4.77; DE = 0.88)]								1.956	(0.052)	0.56
**P-SP** [(girls; $\bar{X}$ = 3.05; DE = 0.08); (boys = 3.25; DE = 0.79)]								0.101	(0.920)	0.52
**P-OD** [(girls; $\bar{X}$ = 4.35; DE = 0.76); (boys = 4.64; DE = 0.66)]								0.018	(0.986)	0.64
**ACP** [(girls; $\bar{X}$ = 3.96; DE = 0.41); (boys = 2.93; DE = 0.43)]								2.357	<0.050 *	0.57
**CA** [(girls; $\bar{X}$ = 1.82; DE = 0.71); (boys = 1.75; DE = 0.73)]								−0.596	(0.552)	0.62
**SCP** [(girls; $\bar{X}$ = 1.60; DE = 1.50); (boys = 3.15; DE = 0.90)]								−4.081	<0.001 **	0.67
**Estimation**	**χ^2^**	**gl**	**CFI**	**RMSEA**	**NCI**	**Δχ^2^**	**ΔCFI**	**ΔRMSEA**		**ΔNCI**
Boys	48.4		0.95	0.042	-	-	-	-		-
Girls	62.3		0.88	0.061	-	-	-	-		-
Unrestricted Model	53.9	27	0.94	0.044	0.098	-	-	-		-
Restricted Model	96.2	33	0.93	0.040	0.999	6.27	0.019	0.004		0.014

Note. * *p* < 0.05; ** *p* < 0.01. d = Cohen coefficient (effect size). SCP: subjective competitiveness perception; P-SO: self-oriented perfectionism; P-SP: socially prescribed perfectionism; P-OD: other-oriented perfectionism; ACP: altruistic prosocial behavior; CA: aggressive behavior.

**Table 2 ijerph-20-02887-t002:** Correlational analysis.

	1	2	3	4	5	6	7	8	9	10
**1. Age**	—									
**2. SCP**	−0.293 **	—								
**3. P-AO**	0.017	0.072	—							
**4. P-SP**	−0.121	0.083	0.424 **	—						
**5. P-OD**	−0.076	0.028	0.178 **	−0.076	—					
**6. ACP**	0.161 **	−0.184 **	−0.072	−0.292 **	−0.209	—				
**7. PB**	0.177 **	−0.128 *	0.187 **	−0.312	−0.206	0.723 ***	—			
**8. PhAB**	−0.289 ***	0.176 **	0.099	0.290 ***	0.297 *	−0.292 ***	−0.475 ***			
**9. VAB**	−0.120	0.003	0.123	0.270 ***	0.355 ***	−0.263 ***	−0.342 ***	0.495 ***		
**10. AB**	−0.281 ***	0.139 *	0.252	0.344 ***	0.365 **	−0.383 ***	−0.521 ***	0.483 ***	0.474 ***	—

Note. * *p* < 0.05; ** *p* < 0.01; *** *p* < 0.00. SCP: subjective competitiveness perception; P-AO: self-oriented perfectionism; P-SP: socially prescribed perfectionism; P-OD: other-oriented perfectionism; ACP: altruistic prosocial behavior; PB: prosocial behavior; VAB: verbally aggressive behavior; PhAB: physically aggressive behavior; AB: aggressive behavior.

## Data Availability

Not applicable.

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
