# Peer review of "Be Prosocial My Friend: The Social Disconnection Model of Perfectionism in Adolescents Immersed in Competitive Sport"

_ijerph, 2023, doi:10.3390/ijerph20042887_

Round 1

Reviewer 1 Report

Dear Authors,

I am pleased to review an original research paper draft "Be prosocial my friend". A good story to improve the perfectionist model of social disconnection in adolescents immersed in competitive contexts". It proposes an interesting theoretical perspective, dataset and interpretations. However, there are several points which can help to significantly improve this work:

1. (2-3) title looks strange. I would suggest entitling the article:

"Be prosocial my friend": the perfectionist model of social disconnection in adolescents immersed in competitive contexts"

2. The abstract is quite informative and structured logically. However, the language style is too dry and technical. It could be easier to digest and reader-friendly. 

3. (89-101) figure 1 should be improved in quality

4. The introduction could be divided into subsections to easier navigate your readers. Since it embraces the theoretical framework as well, it is a bit heavy as a single barely structured piece of complex information.

5. Also, I suggest adding a short paragraph justifying fitting to the journal's scope and the public health field in particular. 

6. I recommend:

Glebova, E. and Desbordes, M., 2020. Technology Enhanced Sports Spectators Customer Experiences: Measuring and Identifying Impact of Mobile Applications on Sports Spectators Customer Experiences. Athens Journal of Sports7(2), pp.115-140.

Barnett, M.D. and Johnson, D.M., 2016. The perfectionism social disconnection model: The mediating role of communication styles. Personality and Individual Differences94, pp.200-205.

7. The first subsection of methods should indicate "data collection" in the subtitle as well

8. What is the data validation procedure?

9. (343) "honestly" should be eliminated since researchers are never dishonest

10. Again, a subsection should be made for limitations and future research directions, to be better recognizable for a reader

11. Please CLEARLY state research implications: theoretical and practical.

Author Response

Dear reviewer 1.

Thank you for your comments and suggestions. We have tried to attend each one of them. We hope you agree with our new version of the manuscript.

  1. (2-3) title looks strange. I would suggest entitling the article: "Be prosocial my friend": the perfectionist model of social disconnection in adolescents immersed in competitive contexts". We have taken your suggestion into account and performed the changes
  2. The abstract is quite informative and structured logically. However, the language style is too dry and technical. It could be easier to digest and reader-friendly. We have attended to your suggestion and have made some changes to the text.
  3. (89-101) figure 1 should be improved in quality. We have created the figure in the better quality possible.
  4. The introduction could be divided into subsections to easier navigate your readers. Since it embraces the theoretical framework as well, it is a bit heavy as a single barely structured piece of complex information. To a better understood and explicative coherence, we have performed different changes to the text (order of the paragraphs and order of cites, subtitles)
  5. Also, I suggest adding a short paragraph justifying fitting to the journal's scope and the public health field in particular. In lines 100-101 (page 2) and 182-194 (page 3) A new paragraph has been included.
  6. I recommend:

-Glebova, E. and Desbordes, M., 2020. Technology Enhanced Sports Spectators Customer Experiences: Measuring and Identifying Impact of Mobile Applications on Sports Spectators Customer Experiences. Athens Journal of Sports7(2), pp.115-140.

-Barnett, M.D. and Johnson, D.M., 2016. The perfectionism social disconnection model: The mediating role of communication styles. Personality and Individual Differences94, pp.200-205.

We have taken your suggestion into account and both references have been added [55;59]

  1. The first subsection of methods should indicate "data collection" in the subtitle as well. We have taken your suggestion into account and added the information
  2. What is the data validation procedure? Lines 294-295 of the manuscript say: “tests of consistency of measurement (Cronbach and CFA) were performed”.
  3. (343) "honestly" should be eliminated since researchers are never dishonest. I agree with you. The consideration has been deleted.
  4. Again, a subsection should be made for limitations and future research directions, to be better recognizable for a reader. The consideration has been added.
  5. Please CLEARLY state research implications: theoretical and practical. We have added a description (lines 555-564) that reinforces the description in lines 594-599)

We have reviewed the references, where there were different dis-adjustment with Vancouver style rules. Now, we believe that each of them is fine.

The authors

Reviewer 2 Report

The authors have touched a very interesting issue in sport and I am glad I am able to assist in improving the quality of their presentation. 

The title is informative and relates to the study. Abstract is well-structured, clearly written and consists of all necessary information in short about the study. I suggest though, changing one of the key words competitive context into competitive sport as the study concerns those young people who practice federated sports. 

Introduction. 

I do not agree with the thesis of the first paragraph stating the 'winning-at-all costs' attitude of the all agents involved in the children and youth sports. 

Even if that is true (according to the provided references) this is mulfunction of youth sport and this should be clearly commented by the authors. Otherwise, it may seem that this is how the authors see the function of youth sport in the stimulation and moral development of young children. 

Another problem is the level of moral competency of the PE teachers and coaches themselves. This has been found in research studies to be overwhelmingly low and that may be an issue as well. Please refer to this as well. 

Also, in Introduction, in Figure 1 first caption I think there is something wrong with the line in the circle and it sounds incorrect, and I think it should go like: "I must be the best, otherwise I will be rejected" - please check with the original. 

Page 3, line 106 starts with a reference number in [24] - this needs to be changed to the name of the author followed by Rice et al. [24]..... 

Generally, in my opinion Introduction is lacking a balance in the rationale - you present and introduce a reader into narrative of the mediating factors taking part in the process of social relationships build-up in youth sport, but very little attention is paid to the young sportspersons-coaches relationships and how powerful and influential role coaches play in inhibiting or stimulating some naturally driven patters of social behaviors. I think, this needs to be tackled more deeply in the Introductory section. 

Methods section

In the first subsection you describe sample of the study and I got a bit confused here. From the abstract I learned that the respondents were coming from the federated sport participants group, while in the text you mention that more girls than boys did not practice any type of physical activity - how come? What were the questions asked about their participation in organized sport? Please expand more on this item. 

Other research instruments have been described in detail and provided with reliability coefficents, which is good. Also references for the original sources were provided. 

In the Results section please advocate and explain the use the Students t-test - I understand there was normal date distribution? because you did not mention.

In Discussion 

Please start the section with brief introduction to your findings. 

Next issue - line 298  the sentence starting from 'It is also interesting to note.....' does not sound grammatically correct, please rephrase.

Also in this section please refer more to the biological maturation processes while explaining occurrence of problems with aggression, and link it to the parallel development of front cortex of the brain in this age (responsible for emotional aspects of behavior). 

I think it would also be adviceable to refer to various sports characteristics (as you have rightfully mentioned in the limitations section), and track the role of individual/team sports on the overall and moral development of young athletes. This is also important factor in analyzing aggressiveness or/and prosocial behaviors. 

Conclusions needs to be build more on the actual findings from your study, and then followed by the recommendations. Please rehrase.

Figures and tables are neat and clear to read. 

References up-to date but this section need to expanded with more sources concerning the topics mentioned in the review. 

Author Response

Dear reviewer 2.

The authors have touched a very interesting issue in sport and I am glad I am able to assist in improving the quality of their presentation. 

The title is informative and relates to the study. Abstract is well-structured, clearly written and consists of all necessary information in short about the study. I suggest though, changing one of the key words competitive context into competitive sport as the study concerns those young people who practice federated sports. We have taken your suggestion into account and performed the changes. Now, the title will be "Be prosocial my friend": the perfectionist model of social disconnection in adolescents immersed in competitive contexts"

Introduction. 

I do not agree with the thesis of the first paragraph stating the 'winning-at-all costs' attitude of the all agents involved in the children and youth sports. We have modified along the text some expressions to also contemplate the positive value of sport and the actions of those around them.

Even if that is true (according to the provided references) this is mulfunction of youth sport and this should be clearly commented by the authors. Otherwise, it may seem that this is how the authors see the function of youth sport in the stimulation and moral development of young children. The detail has been attended to and new references added (lines 40-93).

Another problem is the level of moral competency of the PE teachers and coaches themselves. This has been found in research studies to be overwhelmingly low and that may be an issue as well. Please refer to this as well. The detail has been attended to and new references added (lines 40-93).

Also, in Introduction, in Figure 1 first caption I think there is something wrong with the line in the circle and it sounds incorrect, and I think it should go like: "I must be the best, otherwise I will be rejected" - please check with the original. The detail has been attended to, and the comment in figure 1 has been changed

Page 3, line 106 starts with a reference number in [24] - this needs to be changed to the name of the author followed by Rice et al. [24].....  The detail has been attended to, and the comment has been changed. Now, the reference is [23].

Generally, in my opinion Introduction is lacking a balance in the rationale - you present and introduce a reader into narrative of the mediating factors taking part in the process of social relationships build-up in youth sport, but very little attention is paid to the young sportspersons-coaches relationships and how powerful and influential role coaches play in inhibiting or stimulating some naturally driven patters of social behaviors. I think, this needs to be tackled more deeply in the Introductory section. Thank you for your appreciation. All of the aspects he points out have been incorporated in the arguments of this introduction (reviewer 1 has also assessed some similar ones), adding to his good judgment, contributions and undoubtedly improving the initial description.

Methods section

In the first subsection you describe sample of the study and I got a bit confused here. From the abstract I learned that the respondents were coming from the federated sport participants group, while in the text you mention that more girls than boys did not practice any type of physical activity - how come? What were the questions asked about their participation in organized sport? Please expand more on this item. It was an error in the wording of the data. We have changed the text and removed this date. The questions asked was: federated/nonfederated, type of sport, feeling competitive in the sport and feeling competitive in the school.

Other research instruments have been described in detail and provided with reliability coefficents, which is good. Also references for the original sources were provided. 

In the Results section please advocate and explain the use the Students t-test - I understand there was normal date distribution? because you did not mention. Normal distribution (K-S) and effect size (d Cohen) data was included to complete students t-test information.

In Discussion 

Please start the section with brief introduction to your findings. The detail has been attended to.

Next issue - line 298  the sentence starting from 'It is also interesting to note.....' does not sound grammatically correct, please rephrase. The detail has been attended to and changed.

Also in this section please refer more to the biological maturation processes while explaining occurrence of problems with aggression, and link it to the parallel development of front cortex of the brain in this age (responsible for emotional aspects of behavior). The detail has been attended to and new references added (lines 40-43).

I think it would also be adviceable to refer to various sports characteristics (as you have rightfully mentioned in the limitations section), and track the role of individual/team sports on the overall and moral development of young athletes. This is also important factor in analyzing aggressiveness or/and prosocial behaviors. The detail has been attended to and new references added (lines 35-39).

Conclusions needs to be build more on the actual findings from your study, and then followed by the recommendations. Please rehrase. We have added new comments along the Conclusions (lines 362-377), reinforcing the initial description.

Figures and tables are neat and clear to read. 

References up-to date but this section need to expanded with more sources concerning the topics mentioned in the review. The detail has been attended to and new references added.

We have reviewed the references, where there were different dis-adjustment with Vancouver style rules. Now, we believe that each of them is fine.

The authors

Round 2

Reviewer 1 Report

Dear Authors,

Thank you for your efforts to revise the paper, it has been improved.

However, there are a few minor points:

1. Please add concrete findings and research implications to the abstract

2. References 37 and 72 seem to repeat being too similar

Author Response

Thank you reviewer 2 for your new comments.

  1. Findings and research implications to the abstract have been included.
  2. Reference 72 has been deleted (is the same as reference 37). References 73 and 74 now references 72 and 73.

The changes in the abstract have been highlighted in yellow

We look forward to hearing from you. We remain at your disposal

The authors 
